# Frequency-Based Haze and Rain Removal Network (FHRR-Net) with Deep Convolutional Encoder-Decoder

**Dong Hwan Kim [1], Woo Jin Ahn [2] (ID), Myo Taeg Lim [2,*] (ID), Tae Koo Kang [3,*] (ID) and Dong Won Kim [4] (ID)**

1 Department of Automotive Convergence, Korea University, Seoul 02841, Korea; anwhod@korea.ac.kr
2 Department of Electrical Engineering, Korea University, Seoul 02841, Korea; wjahn@korea.ac.kr
3 Department of Human Intelligence and Robot Engineering, Sangmyung University, Cheonan 31066, Korea
4 Department of Digital Electronics, Inha Technical College, Incheon 22212, Korea; dwnkim@inhatc.ac.kr
* Correspondence: mlim@korea.ac.kr (M.T.L.); tkkang@smu.ac.kr (T.K.K.)

**Abstract:** Removing haze or rain is one of the difficult problems in computer vision applications. On real-world road images, haze and rain often occur together, but traditional methods cannot solve this imaging problem. To address rain and haze problems simultaneously, we present a robust network-based framework consisting of three steps: image decomposition using guided filters, a frequency-based haze and rain removal network (FHRR-Net), and image restoration based on an atmospheric scattering model using predicted transmission maps and predicted rain-removed images. We demonstrate FHRR-Net's capabilities with synthesized and real-world road images. Experimental results show that our trained framework has superior performance on synthesized and real-world road test images compared with state-of-the-art methods. We use PSNR (peak signal-to-noise) and SSIM (structural similarity index) indicators to evaluate our model quantitatively, showing that our methods have the highest PSNR and SSIM values. Furthermore, we demonstrate through experiments that our method is useful in real-world vision applications.

**Keywords:** encoder-decoder network; dilated convolution; image restoration; guided filter; dehaze; derain

## 1. Introduction

Restoring degraded weather images to clean images is vital for many outdoor vision systems, including object detection [1,2] and semantic segmentation [3,4]. Most of these vision systems are designed based on good weather conditions and aim to work well under these conditions. However, in the real world, various weather conditions, such as rain and fog, degrade camera input images and the vision system. Under rainy and foggy conditions, rain streak occludes nearby objects, while fog produces hazy effects with light scattering, reducing the visibility of objects and the performance of vision systems. Figure 1 shows an example of applying an object detection algorithm to a hazy and rainy road image. In the rainy and foggy road image, object recognition error occurs. Figure 1b shows the result of the YOLOv3 algorithm [5]. The first row of Figure 1b shows incorrect recognition of a truck as a vase due to rain and fog. The second row of Figure 1b shows a low image recognition rate due to images degraded by the weather.

Many camera-based advanced driver-assistance systems (e.g., pedestrian detection, lane departure warning, and traffic sign recognition) have been designed for driver convenience. As shown in the above example, rain and haze can cause errors in camera-based systems and cause serious accidents. Therefore, it is essential to study image restoration for road images degraded by rain and fog. However, road images have various characteristics, such as the presence of sky in the image and also a variety of road shapes, road materials, and backgrounds. Due to these characteristics, it is not easy to restore a degraded road image.

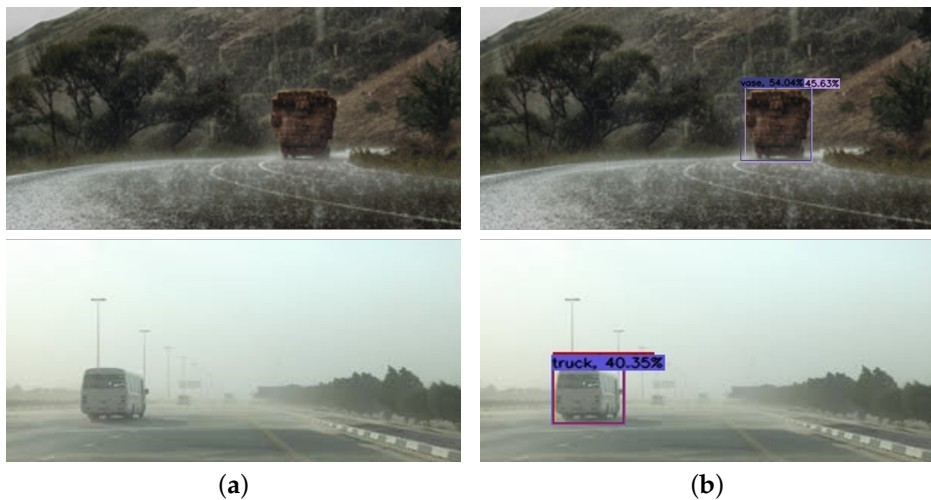

(**a**)             (**b**)

**Figure 1.** Object detection results on haze and rainy road images: (**a**) hazy and rainy input images, (**b**) YOLOv3 [1] object detection results. The images are obtained from Google.

Various methods have been proposed for restoring images degraded by rain and haze. Traditionally, the main focus was on restoration using hand-crafted features. More recently, researchers have mainly studied restoration using neural networks. Rain or haze removal–related studies can be classified as follows:

Rain removal from a single image: Removing rain from a single image has been a challenging task because of the lack of information compared to video-based methods. In spite of such difficulties, many methods using hand-crafted features have been studied. For example, Kim et al. [6] proposed a rain streak removal-based region detection algorithm that analyzes the rotation angle and the aspect ratio of the elliptical kernel at each pixel location, performing non-local mean filtering on detected rain streak regions. Li et al. [7] proposed a simple patch-based priors method using Gaussian mixture models. These priors can accommodate rain streaks at multiple orientations and scales. Du et al. [8] proposed a gradient domain method based on the observation that rain streaks influence X and Y gradients. However, rain removal using only hand-crafted techniques is not completely robust because the problem is ill posed. Recently, deep learning-based methods have been studied to solve this problem. For example, Fu et al. [9] first proposed a deep learning-based rain streak removal method, DerainNet, and upgraded the model [10] by appling ResNet structure to reduce the mapping range, resulting in better convergence of loss functions. Yang et al. [11] proposed a new rain streak image model and removed rain streaks effectively by adding a rain region layer.

Haze removal from a single image: Removing haze from a single image is a difficult problem to solve because it is ill-posed in nature. Recently, however, learning-based methods have been studied, achieving significant progress [12–15]. Zhu et al. [12] proposed a novel color attenuation method, creating a linear model for modeling scene depth. They trained the parameters of the model with a supervised learning method. Cai et al. [13] introduced a CNN-based end-to-end system called Dehazenet. They also introduced a new nonlinear function, BRelu, to increase learning performance.

Ren et al. [14] introduced a deep neural network to estimate transmission maps using haze features effectively. To estimate transmission maps, they proposed a neural network consisting of a layer of large filter sizes and a layer of small filter sizes that fine-tune the results. Li et al. [15] introduced a network called AOD-NET based on a haze model. Instead of estimating the transmission maps and atmospheric light, they combined parameters to create a new model with faster performance than other networks.

In most situations haze and rain occur simultaneously. However, previous methods focused on removing either rain or haze. In other words, these methods are specified with

rain or haze. Yang [11] added a rain region which does not concern about haze models. On the other hand, Ren et al. [14] considers the haze related feature, transmission map.

In this paper, we propose a novel framework for removing rain and haze simultaneously in a single road image. The frequency-based image decomposition is used prior to training neural network models to consider rain and haze separately and present a deep convolution double encoder-decoder neural network model.

## 2. Proposed Frequency-Based Haze and Rain Removal Network (FHRR-Net) Framework

This section introduces methods for analyzing images degraded by rain and fog in detail. We present frequency-based haze and rain removal network (FHRR-Net). Architecture of the network is depicted in Figure 2. Overall framework includes four basic elements:

1.  Analysis of haze and rain models to create a synthesized training dataset.
2.  Image decomposition using a guided filter.
3.  Architecture of encoder-decoder networks.
4.  Image restoration using image degradation model.

In the first section, we illustrate the characteristic of the haze and rain model. Second, we demonstrate a image decomposition method using trainable guided filter. Finally, a deep convolutional encoder-decoder model is introduced which restores hazy and rainy images from degradation model.

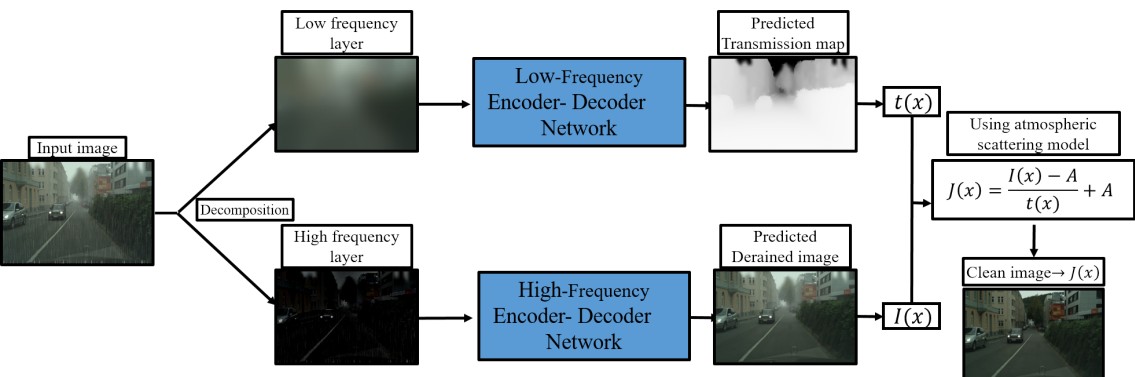

**Figure 2.** Schematic diagram of the proposed frequency-based haze and rain removal network (FHRR-Net) framework.

### 2.1. Analysis of Haze and Rain Models for Creating Synthesized Training Dataset

Training dataset for the proposed network is created based on the rain model as shown in Figure 3. We also propose haze removal method based on the haze model. Therefore, to fully understand FHRR, it is crucial to know the haze and rain model. In this section, we describe the haze model and the rain model in detail.

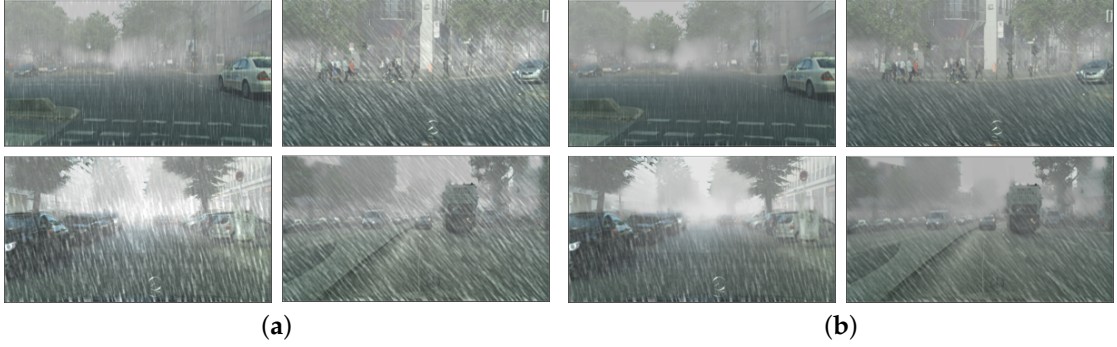

(**a**)             (**b**)

**Figure 3.** Example comparison of synthesis methods on road image using rain-models. (**a**) Using linear addition model, (**b**) Using screen blending model.

2.1.1. Haze Model (Atmospheric Scattering Model)

The physical modeling of haze was done by McCarthy [16] as a atmospheric scattering model. Many traditional hand-crafted base haze removal methods have been studied based on this model [17–19]. The single haze image consists of a combination of haze-free images, transmission maps, and global atmospheric light values. The atmospheric scattering model is written as:

$$I(x) = J(x)t(x) + A(1 - t(x)), \tag{1}$$

where $x$ is a two-dimensional vector that indicates the position of each pixel. $J$ is the clean image, $I$ is the hazy image, $A$ is the global atmospheric light value, and $t$ is the transmission which has a value between 0 and 1. The transmission maps are affected by the scattering coefficient and the distance from the object. The transmission maps is expressed as:

$$t(x) = e^{-\beta d(x)}, \tag{2}$$

where $\beta$ is the scattering coefficient. $d$ indicates distance from the object and the observer in pixel $x$. As the distance of the object and the atmospheric degradation factor increase, the atmosphere becomes further hazy.

2.1.2. Rain Model (Rain Streak Model)

In rainy road environments, the input image is usually affected by rain streak. Thus, the rain streak model is appropriate for representing rainy images. The rain model used to synthesize the shape of the rain streaks, which has been widely used in previous studies, is as follows:

$$O(x) = B(x) + S(x), \tag{3}$$

where $O$ is the observed image degraded by the raindrop. $B$ indicates the clean background image, and $S$ is the rain streak image. There are many studies on removing rain using this model [20,21]. They focused on separating background $B$ and rain $S$ through observed $O$. However, this simple addition model is insufficient to represent the rain phenomenon. For this reason, Luo et al. [22] suggested using a nonlinear composite model with a screen blending method rather than using the existing linear addition synthesis model. The screen blend model can model some visual properties of real-world rain images, such as the effectiveness of internal reflections, to generate visually more true rain images. The combination of rain and background layers can more naturally express signal-defendant properties using screen blending techniques. Therefore, the screen blend model is expected to be more effective in training the network by expressing the combination of background and background more naturally. The model proposed by Luo [22] is as follows:

$$O(x) = B(x) + S(x) - B(x) \circ S(x), \tag{4}$$

where $\circ$ operator indicates point-wise multiplication. This nonlinear synthesis models was used for the training dataset.

2.1.3. Heavy Rain Model

In this section, the final model of the combined haze model and rain model is written as:

$$O(x) = I(x) + S(x) - I(x) \circ S(x), \tag{5}$$

where $I$ means the haze image as expressed in (1). $S$ is the rain streak image as described in (3). Through the above analysis, model rain streaks and haze on images can be expressed mathematically. Therefore, training dataset created on the blending model presents more realistic haze and rain phenomena. The following subsection describes how to decompose the degraded images into frequency domain which helps the network optimize more quickly and accurately.

### 2.2. Image Decomposition Using Guided Filter

One of the most effective edge preserving smoothing algorithms is the guided filter [23,24]. We apply a guided filter to the degraded input image by haze and rain and then obtain the low-frequency layer of the degraded image. The difference between the obtained low-frequency layer and the input image is used to obtain a high-frequency layer of the degraded image. As shown in Figure 4, the high-frequency layer contains most of the rains streaks and edge components. The low-frequency layer looks very blurry, but some elements present in the low-frequency layer constitute a degraded input image excluding rain streaks and edge components present in the high-frequency layer. Therefore, it is possible to deduce that the components of the haze exist in the low-frequency layer. Figure 4 shows an example of the applying guide filters to degraded input images. Figure 4c shows that there are many raindrops and edge components. We use decomposed layers to restore rain and haze images.

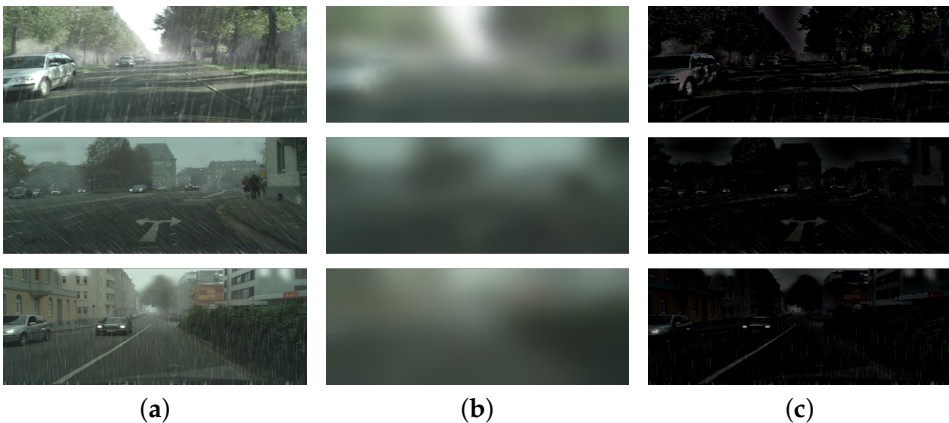

**Figure 4.** Applying guided filter on the hazy and rainy input image. (**a**) Input image. (**b**) Low-frequency of the image. (**c**) High-frequency of the image.

Image decomposition using a guided filter has been proved to be easier to remove rain streaks and achieve excellent results [9,10]. Proposed FHHR-Net estimates transmission maps using only low-frequency layers, which are more effective than using whole single image. Since the block operation is used for convolution, the estimated transmission maps have block artifacts. Because block artifacts adversely affect the dehaze process, various methods have been proposed to eliminate block artifacts [25]. However, when estimating the transmission maps using a low-frequency image, block artifacts are rarely generated because the edges are already smoothed.

In this work, to take advantage of end-to-end learning, we use trainable guided filter [26]. The image decomposition process using guided filter is as follows:

$$O(x) = O_{low}(x) + O_{high}(x), \tag{6}$$

where $O$ indicates hazy and rainy image, which is separated in low-frequency $O_{low}$ and high-frequency $O_{high}$. FHRR-Net uses $O_{high}$ to remove non-stripes and $O_{low}$ to estimate the transmission maps effectively.

### 2.3. Frequency-Based Haze and Rain Removal Network (FHRR-Net) Architecture

Restoring degraded images by haze and rain using the models is equivalent to solving the inverse problem of the models. Deep neural network is used to address the lack of information problem that is characteristic of a single degraded image. In this section, details of the structure of proposed encoder-decoder networks is introduced. The details of the overall network architecture are shown in Table 1.

**Table 1.** FHRR-Net architecture details.

| High-Frequency Encoder-Decoder Network | | | | |
|---|---|---|---|---|
| **Layer** | **Kernel Size** | **Dilated Factor** | **Output Size** | **Skip Connections** |
| Input | | | $256 \times 256$ | |
| Encoder_conv 0 | 3 | 1 | $16 \times 256 \times 256$ | |
| Encoder_conv 1_1 | 3 | 2 | $32 \times 128 \times 128$ | |
| Encoder_conv 1_2 | 3 | 2 | $32 \times 128 \times 128$ | Decoder_deconv 2 |
| Encoder_conv 2_1 | 3 | 2 | $64 \times 64 \times 64$ | |
| Encoder_conv 2_2 | 3 | 2 | $64 \times 64 \times 64$ | Decoder_deconv 1 |
| Encoder_conv 3_1 | 3 | 2 | $128 \times 32 \times 32$ | |
| Encoder_conv 3_2 | 3 | 2 | $128 \times 32 \times 32$ | |
| Decoder_deconv 1 | 2 | 2 | $64 \times 64 \times 64$ | Encoder_conv 2_2 |
| Decoder_conv 1_1 | 3 | 2 | $64 \times 64 \times 64$ | |
| Decoder_conv 1_2 | 3 | 2 | $64 \times 64 \times 64$ | |
| Decoder_deconv 2 | 2 | 2 | $32 \times 128 \times 128$ | Encoder_conv 1_2 |
| Decoder_conv 2_1 | 3 | 2 | $32 \times 128 \times 128$ | |
| Decoder_conv 2_2 | 3 | 2 | $32 \times 128 \times 128$ | |
| Decoder_deconv 3 | 2 | 2 | $32 \times 256 \times 256$ | |
| Decoder_conv 3_1 | 3 | 2 | $16 \times 256 \times 256$ | |
| Decoder_conv 3_2 | 3 | 2 | $16 \times 256 \times 256$ | |
| Output | 3 | 2 | $3 \times 256 \times 256$ | |
| Low-frequency Encoder-decoder network | | | | |
| Layer | Kernel size | Dilated factor | Output size | Skip connections |
| Input | | | $256 \times 256$ | |
| Encoder_conv 0 | 3 | 1 | $16 \times 256 \times 256$ | |
| Encoder_conv 1_1 | 3 | 2 | $32 \times 128 \times 128$ | |
| Encoder_conv 1_2 | 3 | 2 | $32 \times 128 \times 128$ | Decoder_deconv 2 |
| Encoder_conv 2_1 | 3 | 2 | $64 \times 64 \times 64$ | |
| Encoder_conv 2_2 | 3 | 2 | $64 \times 64 \times 64$ | Decoder_deconv 1 |
| Encoder_conv 3_1 | 3 | 2 | $128 \times 32 \times 32$ | |
| Encoder_conv 3_2 | 3 | 2 | $128 \times 32 \times 32$ | |
| Decoder_deconv 1 | 2 | 2 | $64 \times 64 \times 64$ | Encoder_conv 2_2 |
| Decoder_conv 1_1 | 3 | 2 | $64 \times 64 \times 64$ | |
| Decoder_conv 1_2 | 3 | 2 | $64 \times 64 \times 64$ | |
| Decoder_deconv 2 | 2 | 2 | $32 \times 128 \times 128$ | Encoder_conv 1_2 |
| Decoder_conv 2_1 | 3 | 2 | $32 \times 128 \times 128$ | |
| Decoder_conv 2_2 | 3 | 2 | $32 \times 128 \times 128$ | |
| Decoder_deconv 3 | 2 | 2 | $32 \times 256 \times 256$ | |
| Decoder_conv 3_1 | 3 | 2 | $16 \times 256 \times 256$ | |
| Decoder_conv 3_2 | 3 | 2 | $16 \times 256 \times 256$ | |
| Output | 3 | 2 | $1 \times 256 \times 256$ | |

Inspired by the residual network [27] and the encoder-decoder style network [28], the proposed FHRR-Net consists of two encoder-decoder models: one is for training the high-frequency layer of the input image, and one is for training the low-frequency layer of the image. Dilated structures and symmetric skip connections are applied to increase the image restoration performance of the network as illustrated in Figure 5. It also shows the linking of the encoder and decoders through symmetric skip connections. After the image is decomposed into low and high frequency domain using guided filter, it goes through high-frequency network or low-frequency network. Both network architectures are symmetric, which is efficient for estimating frequency domain into image domain.

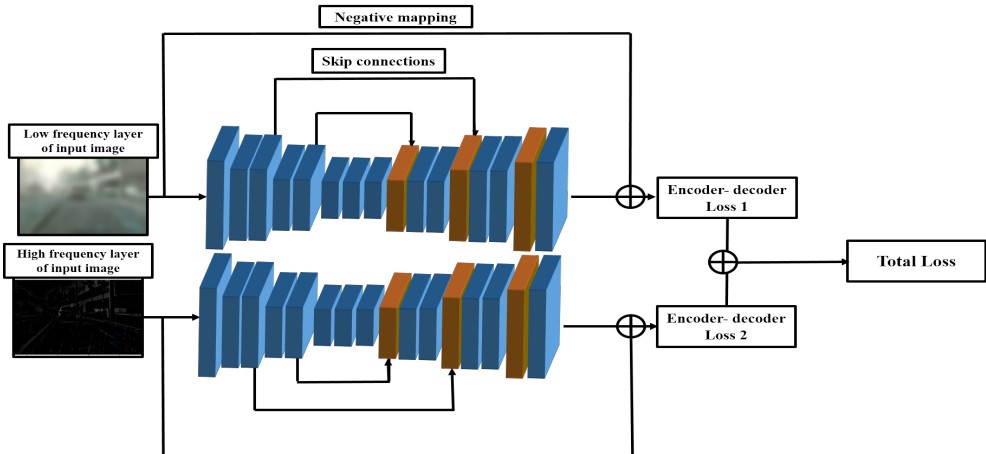

**Figure 5.** The overall architecture of FHRR-Net. The encoder decoder structure consists of a convolutional layer, followed by a batch normalization layer and a ReLu layer. The blue boxes indicate convolutional layers, and orange boxes indicate deconvolutional layers.

### 2.3.1. Dilated Convolution

Dilated Convolution [29,30] is a way to increase the receptive field by adding zero padding inside the filter without increasing the parameters. Dilation rate is a hyper parameter for dilated convolution which defines the spacing between kernels. For example, a $3 \times 3$ kernel with dilation rate of 2 has the same view as a $5 \times 5$ kernel, using only nine parameters. A visual example of dilated convolution is shown in Figure 6.

In Figure 6, $3 \times 3$ dilated convolution with a dilated rate of 1 has the same receptive field as a traditional $3 \times 3$ convolution, but a $3 \times 3$ dilated convolution with a dilated rate of 2 has a receptive field of $5 \times 5$ convolution. Figure 6b shows the same amount of computation as the traditional $3 \times 3$ convolution, but with a $5 \times 5$ receptive field.

In the haze removal work, contextual information of the input image is helpful for removing haze relevant features [14] and also useful in rain streak removal work [11]. By using a dilated structure, richer contextual information from the input can be obtained.

Dilated convolutions for all the convolutional layers of proposed encoder-decoder structure and also, due to the nature of the encoder-decoder structure, it has a variety of scale output features. These characteristics make it possible to take a large receptive field and easily obtain contextual information of the input image.

### 2.3.2. Symmetric Skip Connections and Negative Mapping

The encoder-decoder network with symmetric skip connections are suitable to restore degraded image [27]. The encoder of proposed network consists of convolutional layers, and the decoder consists of deconvolutional and convolutional layers. The encoder works to remove the major noise while preserving the important characteristics of the image. As the network goes deeper, the decoder works less because details along with the main noise can be removed. Symmetric skip connections have been proposed to solve this problem in deep networks which deliver the details of the image from the encoder to the decoder, increasing the image's restoring performance. To increase the training ability of the network, we also applied Fu [10]'s negative mapping method which is one of the skip connections. The negative mapping technique is implemented in the network to help training by reducing the mapping range as shown in Figure 5. Using negative mapping significantly reduces the mapping range. Therefore, we suggested applying a negative mapping method to the encoder-decoder network to restore the detail of the image better.

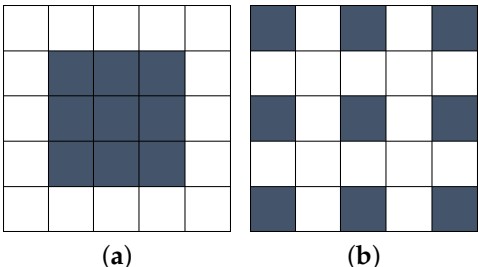

**Figure 6.** A visual example of dilated convolution. (**a**) Traditional convolution. (**b**) Dilated convolution.

2.3.3. Loss Functions for Training

Since we estimate the output through two encoder-decoder networks, we need to define two loss functions for high-frequency and low-frequency network. L1 error is known to produce better results in image restoration than L2 loss. This phenomenon is due to the convergence problem characteristics: L2 tends to converge into to the local minimum, but L1 converges well to the global minimum in image restoration [31]. Therefore, we train each encoder-decoder network using L1 errors to get better image restoration results from training dataset. Based on the introduced statement, the loss function of the high-frequency encoder-decoder is as follows:

$$L_H = \sum_{x \in X} |F_H(O_{high}(x)) + O(x) - I(x)|, \tag{7}$$

where $x$ denotes the pixel in image spatial domain $X$, and $F_H$ denotes proposed high frequency encoder-decoder network. $O_{high}$ is the high-frequency region of the input image which is obtained from guided filter as mentioned. $I$ denotes a ground truth image where only haze exists. Through the loss function, the high-frequency encoder-decoder network trains and outputs rain-removed images from the haze and rain images.

Second, the loss function of the low-frequency encoder-decoder is as follows:

$$L_L = \sum_{x \in X} |F_L(O_{low}(x)) + O(x) - t(x)|, \tag{8}$$

where $t$ denotes transmission map of the input image, $O_{low}$ is the low-frequency region of the input images, and $F_L$ denotes proposed low frequency encoder-decoder network. Unlike the high-frequency encoder-decoder network, the ground truth of the low-frequency loss function is the transmission map. The other parameters of the loss function are the same as the parameters of the high-frequency loss function.

Since the two-loss functions has same scale, the total loss is a made with addition operation. The total loss function can be expressed as follows:

$$L_T = \alpha \cdot L_L + (1 - \alpha) \cdot L_H, \tag{9}$$

where $L_T$ is the total loss function. The network is trained through optimizing function in (9). We empirically set the alpha to 0.7 for training and design our network structure, as described throughout this section.

*2.4. Image Restoration Using Atmospheric Scattering Model*

In this section, we describe the final process of the framework. After estimating the transmission map from the low-frequency network and removing the rain streaks with the high-frequency network, using these information, degraded images are restored based on physical methods by rewriting the atmospheric model 1 into following equation:

$$J(x) = \frac{I(x) - A}{t(x)} + A. \tag{10}$$

Throughout proposed networks, estimated transmission maps and estimated rain-removed images are obtained. Specifically, we can obtain $t$ and $I$ using the low-frequency high-frequency encoder-decoder networks, respectively. The value of $t$ can be zero value in an image, such as bright light and sky regions. In these pixels, the value in (1) can have the infinite value. Therefore, a lower limit of $t(x)$ was set to 0.1 to prevent the denominator from zero. Additionally, we need to estimate the atmospheric light $A$ based on the following equation:

$$I(x) = A, \ if, \ t(x) < t_{th}. \tag{11}$$

In (11), we choose the darkest 0.1% of pixels in a transmission map $t(x)$. This part means the highest intensity haze in the hazy image $I(x)$, as the atmospheric light $A$. In order to set the threshold for $A$, a threshold value experiment is needed, which we perform in the next section. As above, $I$, $t$, and $A$ are determined. We assign values to the image degradation model to get a clean image.

## 3. Experimental Result

In this section, extensive experiments are conducted to demonstrate the performance of proposed FHRR-Net. Results on real-world road image and synthetic road data are used to evaluate the performance of our model against several state-of-the-art neural network methods. The synthetic dataset restoration score is quantitatively measured by PSNR (peak signal-to-noise) and SSIM (structural similarity index) which are often used as indicators for evaluating performance in several image restoration works. Real-world road image experiment results are also compared qualitatively. We also show the effectiveness of proposed method through ablation study.

### 3.1. Experimental Environment

We conducted experiments on Cityscapes' foggy dataset [32–34], which is widely used for other research and shows meaningful results on real world dataset. It provides a collection of synthetic foggy images and transmission maps (ground-truth material). Using foggy dataset and rain streaks [10] with (5), we collected 18,006 training images, 2871 validation images, and 9189 test images. In addition, using models pretrained on generated Cityscapes rain dataset, 24 real world road images were tested which includes haze and rain streak effects. FHRR-Net and other comparison methods were trained and tested on same dataset condition. After each convolutional layer in the encoder-decoder network, batch normalization [35] and ReLu [36] layer are used. Adam optimizer method [37] was used to train the network with batch size 128, momentum to 0.9. Initial learning rate is set to 0.01 for 100K iterations and 0.001 for 100K iterations to 200K iterations and terminate the training at 200K iterations. All experiments are conducted using NVIDIA Geforce GTX 1080 with Tensorflow library.

### 3.2. Creating the Synthesized Dataset

As mentioned above, new rain and haze road dataset are generated using (5). Rain streaks are applied on Cityscapes' foggy dataset. Refs. [32–34] provides a collection of synthetic foggy images and transmission maps, which is ground-truth for (8). The fog data consisted of images of the vehicle front camera in various cities in conditions ranging from light to dense fog, as reflected in various scattering values $(0.02, 0.01, 0.005)$. The synthesized dataset expresses the harsh road situation, which includes not only rain and haze on the road but also haze caused by heavy rain on the road. Figure 7 shows a visual representation of the rain and haze road dataset created by combining the Cityscapes foggy dataset with the rain model.

This synthesized rain and haze dataset is used as input to the proposed FHRR-Net. To train the network, transmission maps in the foggy dataset is set to the ground truth of the low-frequency network and the hazy image in the foggy dataset is set to the ground truth of the high-frequency network. The loss function is optimized by comparing the ground truth of the dataset with the network output of the FHRR-Net.

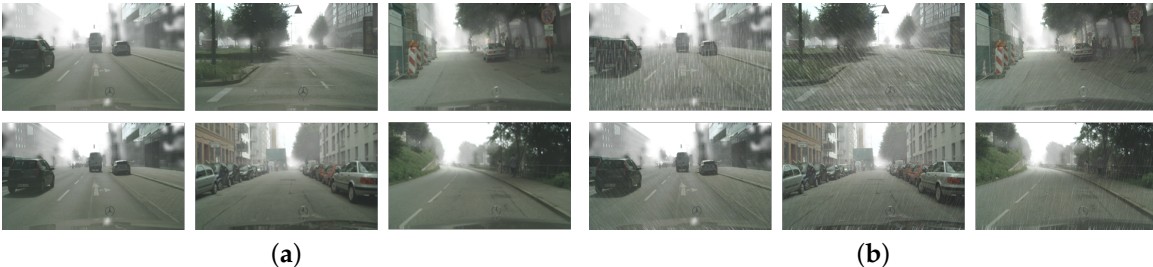

Figure 7. Example of created synthetic dataset. (**a**) Cityscape foggy dataset [32–34]. (**b**) Our new rain and haze road dataset. To make the dataset more general, we created a new rain and haze road dataset by combining various directions and thicknesses of the rain streak.

### 3.3. Performance Comparison with State-of-the-Art Methods

A comparative experiment was conducted with the proposed FHRR-Net and two state-of-art methods: MSCNN [14] and DerainNet [10]. We first compare the state-of-the-art methods on synthetic datasets, evaluating qualitatively and quantitatively. Figure 8b is a part of the synthesized test data which includes both rain and haze components. We also compare the state-of-the-art methods on real-world road images to show that the network is robust.

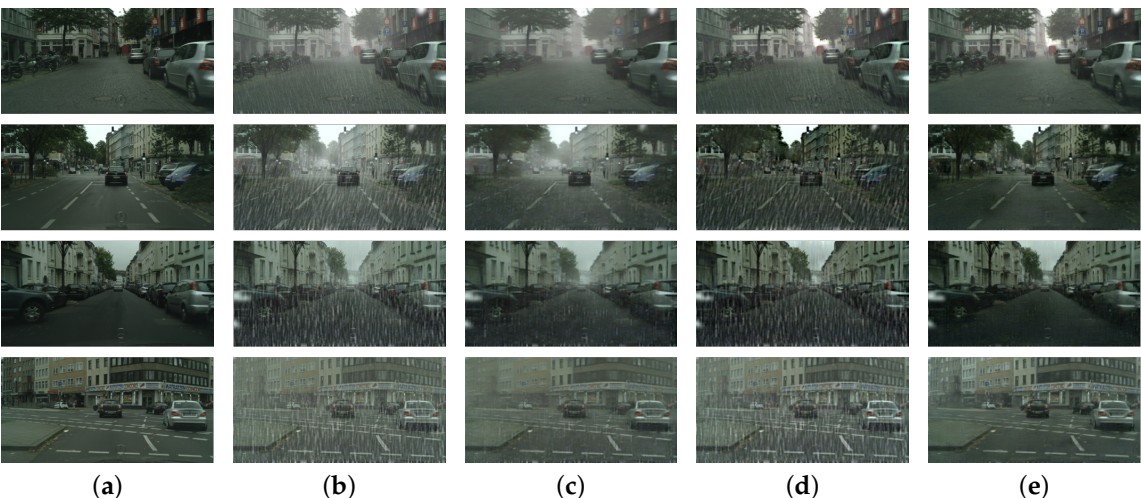

Figure 8. Visual comparison with the state-of-art methods on the synthesized dataset. (**a**) Ground truth, (**b**) Inputs, (**c**) DerainNet [10] (**d**) MSCNN [14], (**e**) FHRR-Net.

In Figure 8, DerainNet only removes the rain streak part of the synthesized image. Specifically, Figure 8c shows that the fog is not removed. On the other hand, MSCNN only removes the hazy part of the synthesized image. In Figure 8d, the rain streak components are not removed with MSCNN. Quantitative evaluations of the results on Figure 8 are shown in Table 2. FHRR-Net obtains better PSNR and SSIM indicators than other state-of-the-art methods. Based on the above experimental results, FHRR-Net shows the best results for the synthesized data.

Visual comparison results from real-world road data are shown in Figure 9. Some images on Figure 9c show that MSCNN is unable to remove rain from the actual road image with haze and rain together. In addition, there is excessive enhancement in the process of removing the haze.

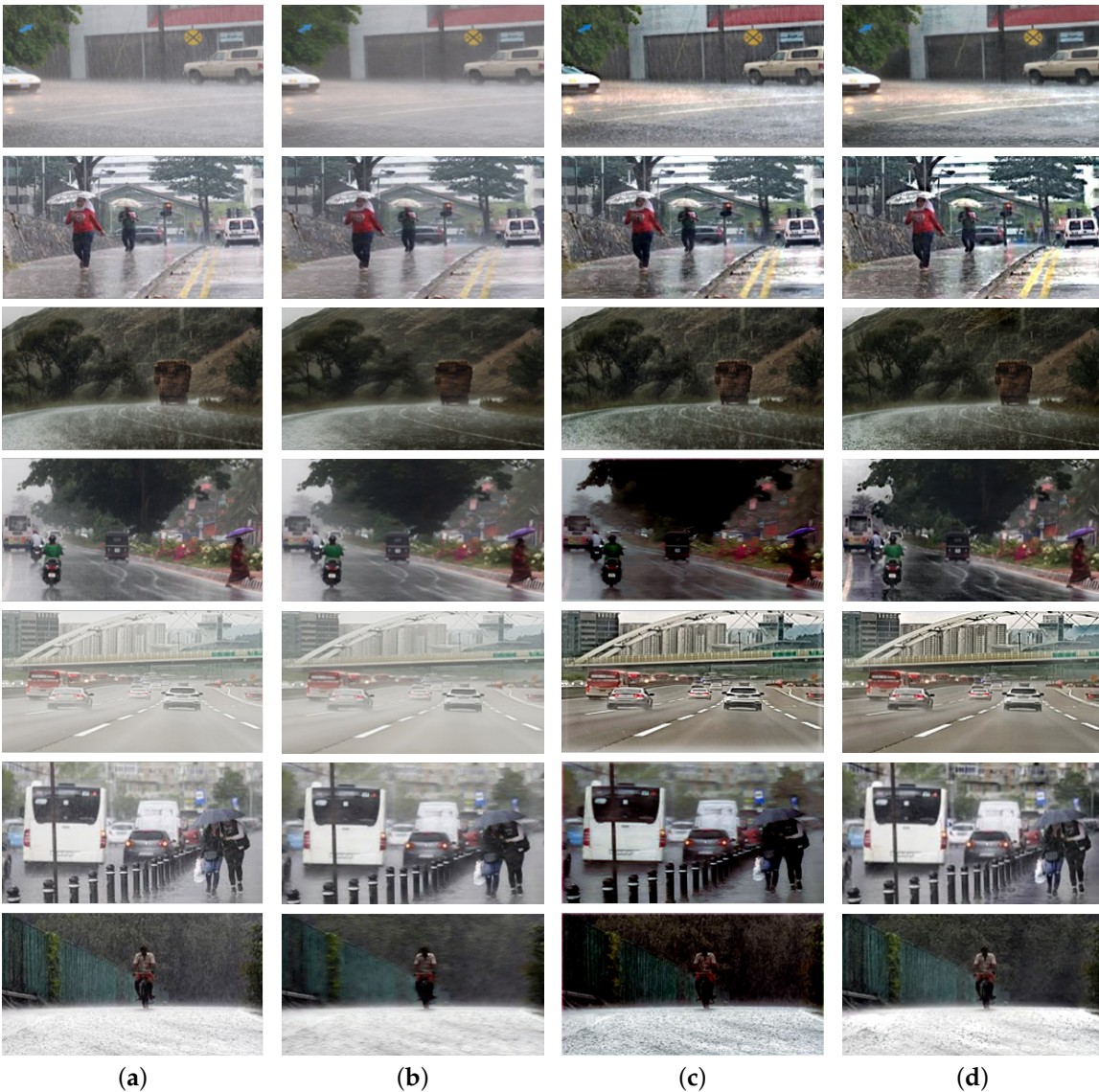

      (**a**)            (**b**)            (**c**)            (**d**)

**Figure 9.** Visual comparison with the state-of-art methods on real-world road data, (**a**) Inputs, (**b**) DerainNet [10] (**c**) MSCNN [14], (**d**) FHRR-Net. Some images are obtained through Google Images; others are taken directly.

**Table 2.** Average PSNR (peak signal-to-noise) and SSIM (structural similarity index) results on synthesized data for state-of-arts methods.

| Methods | Average PSNR | Average SSIM |
|---|---|---|
| DerainNet | $\mu = 18.79, \sigma^2 = 1.58$ | $\mu = 0.84, \sigma^2 = 0.08$ |
| MSCNN | $\mu = 20.01, \sigma^2 = 1.08$ | $\mu = 0.81, \sigma^2 = 0.08$ |
| FHRR-Net | $\mu = 22.02, \sigma^2 = 1.20$ | $\mu = 0.91, \sigma^2 = 0.06$ |

*3.4. Comparison of Performance with Combined SOTA Methods (Derain + Dehaze)*

    In this section, we conducted an experiment comparing the proposed method with the results of applying the state-of-the-art techniques DerainNet [10] and MSCNN [14] sequentially. The experiment is described in the following, Figure 10, schematic diagram.

    Figure 11 shows the results of an experiment performed as described in Figure 10 schematic diagram. Experiment 2 method result image usually appears to have excessive color enhancements. For example, in the first picture of Figure 11c,d, the color of the tree on the upper left is close to black. In addition, the color of the people in the middle picture

is also close to black.Person, car, tree, or umbrella in the last picture is shown close to black. As a result, in the Figure 10, experiment 2 and experiment 3 methods excessively improve the color compared to the original image. Hence, this experiment confirm that FHRR-Net improves better than the combination of the two SOTA methods in both removing rain and haze.

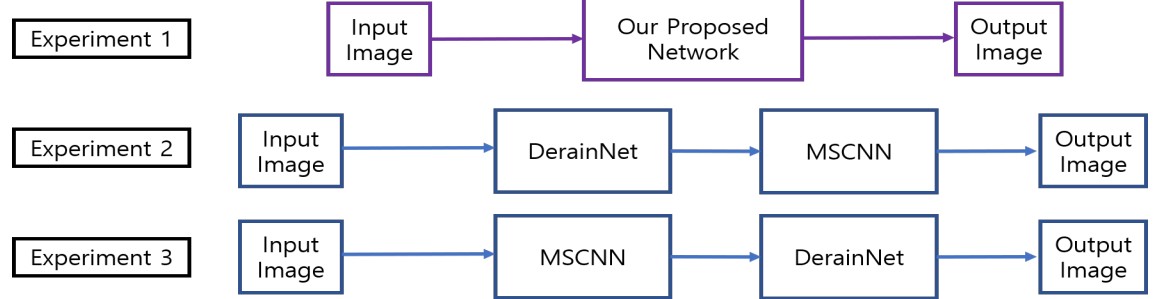

**Figure 10.** Schematic diagram: Comparison experiments with combined SOTA methods and proposed algorithm.

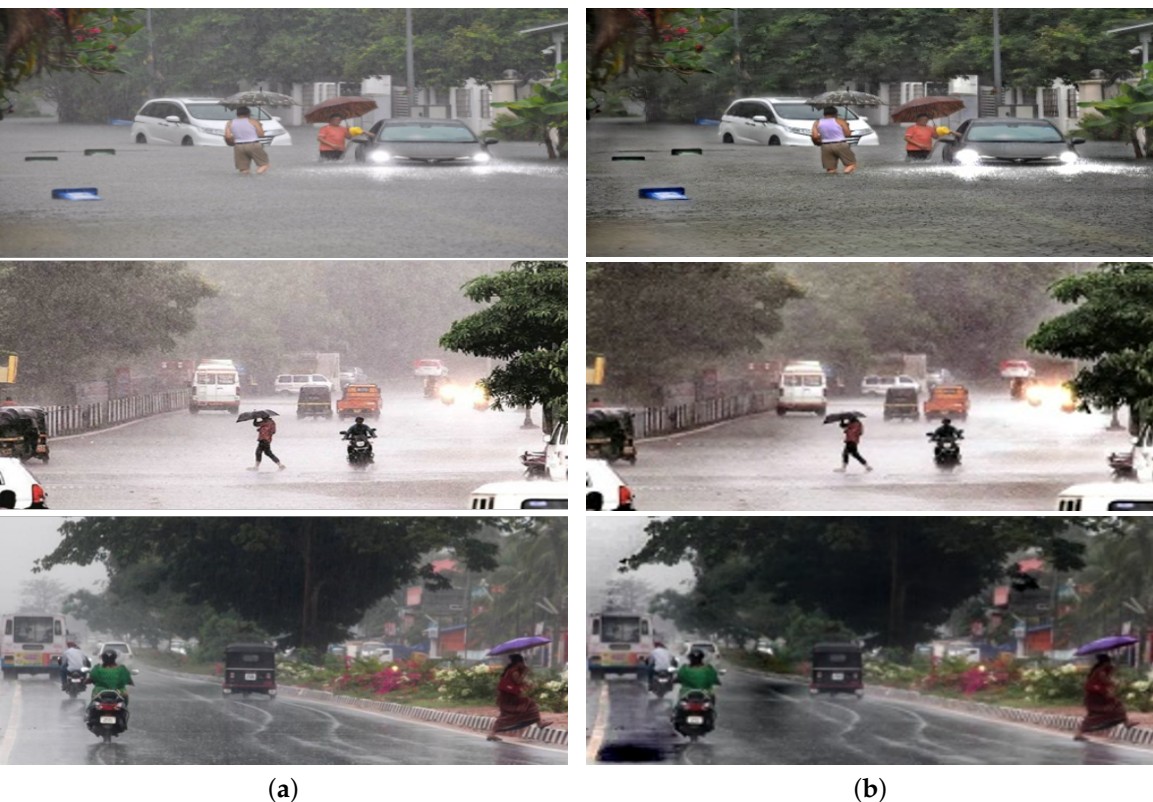

(**a**)          (**b**)

**Figure 11.** *Cont.*

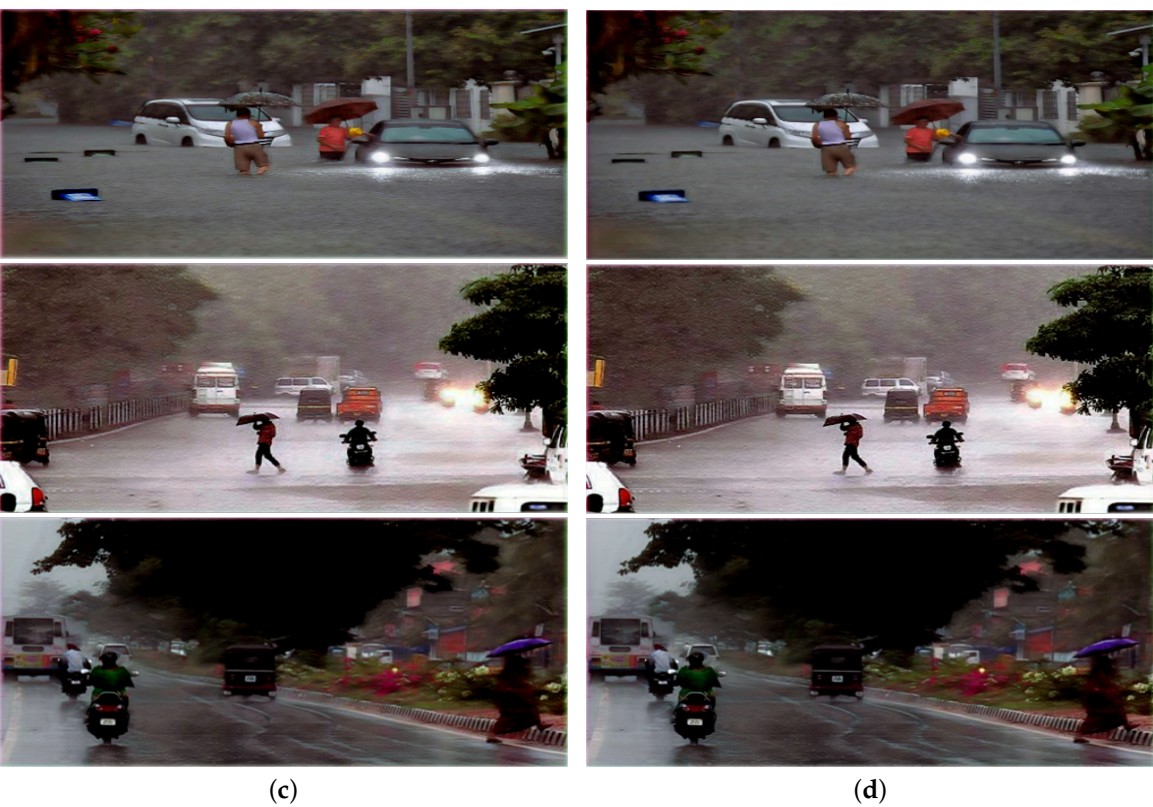

<div align="center">(<b>c</b>)              (<b>d</b>)</div>

**Figure 11.** Comparison results of the combined SOTA methods with FHRR-Net. (**a**) Ground truth. (**b**) Our results. (**c**) Results of applying the (DerainNet + MSCNN) method. (**d**) Results of applying the (MSCNN + DerainNet) method.

### 3.5. Object Detection Results

In order to provide the evidence benefits on harsh road scenes, an experiment is done to prove whether our method affects the results of object detection. YOLOv3 [5], an object detection method, is used for detection which is trained with COCO dataset for the experiments.

Figure 12a shows object detection results on degaded input images, where (b) shows object detection results after applying DerainNet on input images, (c) shows object detection results after applying MSCNN on input images, and (d) shows object detection results after applying FHRR-Net. The first row of Figure 12a shows that object detection fails in the case of degraded images of distant vehicles. In contrast, the first row of Figure 12d shows that even distant vehicles can be detected as objects. Figure 12b,c show the object detection results of DerainNet and MSCNN: object detection results are better than in Figure 12b,c, the object detection result is better than Figure 12a, but the overall accuracy is lower than in Figure 12c.

Table 3 shows detection accuracy results after using state-of-the-art image restoration methods. We use 20 real-world test images to measure and compare average mAP and confidence for each object in the images. It is shown that our algorithm positively affects object detection in degraded images via Table 3 and Figure 12.

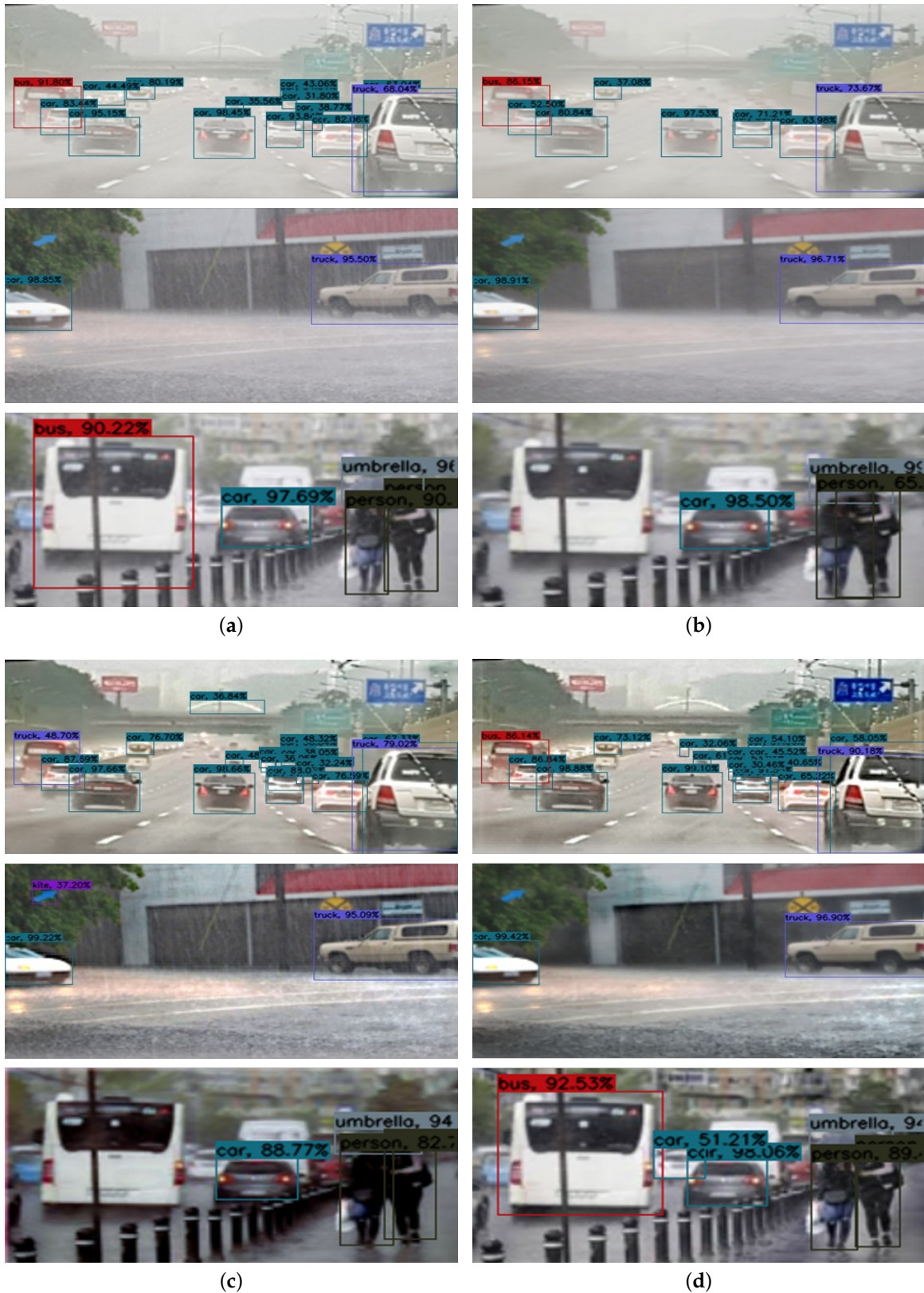

**Figure 12.** YOLOv3 object detection results on real-world road restored images. (**a**) Input images. (**b**) Object detection results after applying DerainNet. (**c**) Object detection results after applying MSCNN. (**d**) Object detection results after applying FHRR-Net.

**Table 3.** Comparison of YOLOv3 results after using the state-of-arts image restoration methods.

| Original Input + YOLOv3 | | DerainNet +YOLOv3 | | MSCNN +YOLOv3 | | FHRR-Net +YOLOv3 | |
|---|---|---|---|---|---|---|---|
| mAP | Confidence | mAP | Confidence | mAP | Confidence | mAP | Confidence |
| 0.84 | 0.924 | 0.77 | 0.909 | 0.75 | 0.845 | 0.89 | 0.921 |

### 3.6. Analysis of Proposed Architecture

3.6.1. Effectiveness of the Proposed Dilated Convolution and Negative Mapping

In Section 2.3, FHRR-Net uses dilated convolution with negative mapping for convolution. To prove that the proposed structure is most effective, we conducted a comparative experiment after excluding the dilated convolution and the negative mapping structure, respectively. In Table 4, quantitative measurement results are shown using PSNR and SSIM scores for each structure on the synthesized dataset. Figure 13 also shows a visual comparison of qualitative results for a synthesized dataset.

Figure 13a shows ground truth of experiments. Figure 13b shows our synthesized test dataset. Figure 13c illustrates a structure learned using a negative mapping structure without using dilated convolution. Figure 13d shows a structure learning using only dilated convolution without using neg mapping structure. The fog is slightly removed as in Figure 13c than in Figure 13e. On the other hand, comparing Figure 13d with Figure 13e shows that the rain streak is less removed without negative mapping. These results implies that removing haze is effected by dilated convolution which gives large receptive field. In addition, it proves that negative mapping gives high-level features through the connection. Table 4 also shows that PSNR and SSIM index are the highest in the proposed structure that uses dilated convention and negative mapping.

As a result of quantitative and qualitative evaluation, dilated convolution and negative mapping are shown to be effective in removing haze and rain from encoder-decoder networks for image restoration.

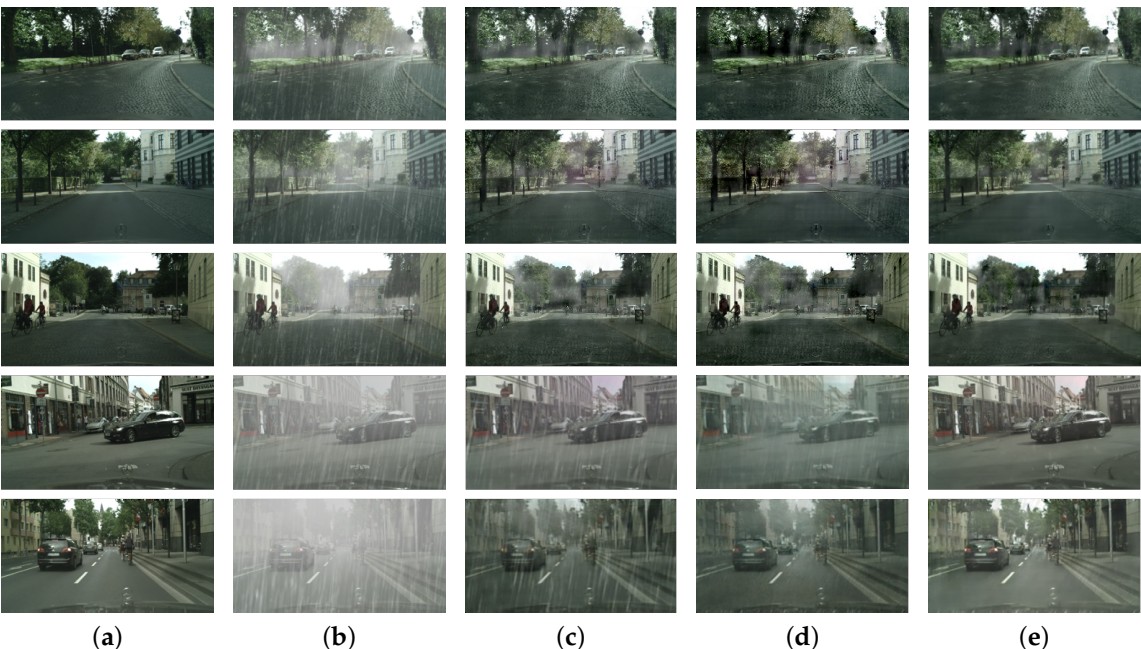

    (**a**)       (**b**)       (**c**)       (**d**)       (**e**)

**Figure 13.** Comparison with different structures on the synthesized dataset, (**a**) Ground truth. (**b**) Synthesized data. (**c**) Structure without dilated convolution. (**d**) Structure without negative mapping. (**e**) FHRR-Net.

### 3.6.2. Comparison of Restoration Performance for Various Threshold Values

In this section, we quantitatively compare the restoration results for each A value determined by each threshold in the image restoration process. Table 5 demonstrates the quantitative measurement results using PSNR and SSIM scores for each threshold value on the test dataset.

As mentioned in Section 2.4, a haze removal method using the image degradation model through transmission maps needs the global atmospheric light to calculate the clean image. In (11), the atmospheric value is determined by selecting average of the 0.1% darkest pixel in the transmission map.

As shown in Table 5, the best results on PSNR and SSIM values are obtained when choosing average of the 0.1% darkest pixel in the transmission map.

Rain and haze are significant obstacles to operating many camera-based vision systems because of the degraded input image. In real-world road conditions, not only good weather but also hazy and rainy weather exist, which can cause serious visual problems. Many studies have been conducted on removing rain and haze from images, but none have studied removing rain and haze simultaneously

**Table 4.** Average PSNR and SSIM with different structures on the synthesized dataset.

|  | Without Dilated Convolution | Without Negative Mapping | FHRR-Net |
|---|---|---|---|
| Average PSNR | $\mu = 21.582, \sigma^2 = 1.15$ | $\mu = 21.96, \sigma^2 = 1.23$ | $\mu = 22.02, \sigma^2 = 1.20$ |
| Average SSIM | $\mu = 0.88, \sigma^2 = 0.06$ | $\mu = 0.89, \sigma^2 = 0.07$ | $\mu = 0.91, \sigma^2 = 0.06$ |

**Table 5.** Average PSNR and SSIM for each threshold to determine A.

|  | Darkest 0.05% Pixel | Darkest 0.1% Pixel | Darkest 0.15% Pixel | Darkest 0.2% Pixel |
|---|---|---|---|---|
| Average PSNR | $\mu = 21.96, \sigma^2 = 1.21$ | $\mu = 22.02, \sigma^2 = 1.22$ | $\mu = 22.00, \sigma^2 = 1.23$ | $\mu = 21.94, \sigma^2 = 1.20$ |
| Average SSIM | $\mu = 0.9105, \sigma^2 = 0.0617$ | $\mu = 0.9105, \sigma^2 = 0.0616$ | $\mu = 0.9105, \sigma^2 = 0.0617$ | $\mu = 0.9103, \sigma^2 = 0.0612$ |

### 3.6.3. Effectiveness of Frequency Decomposition

This section shows the effectiveness of using decomposed image for training the network. In Section 2.3, we proposed a frequency-based framework to improve degraded images using guided filters. As mentioned before, FHRR-Net has the advantage that parameter optimization is easier than learning immediately by the original input. This allows to take faster learning times and obtain better improved images.

Experiment on test dataset implies the effect of preprocessing in Table 6. One is learned without frequency-based image decomposition which only takes the raw image and the second is proposed frequency-based method. Table 6 shows experimental results for two methods. As seen in the table, preprocessing gives an easier optimization for the network.

**Table 6.** Experimental results on frequency-based method effects.

| Methods | Average PSNR | Average SSIM |
|---|---|---|
| Without frequency-based method | $\mu = 21.02, \sigma^2 = 1.55$ | $\mu = 0.82, \sigma^2 = 0.08$ |
| FHRR-Net | $\mu = 22.02, \sigma^2 = 1.20$ | $\mu = 0.91, \sigma^2 = 0.06$ |

### 4. Conclusions

Rain and haze are significant obstacles to camera-based vision systems because of the degraded input image. In real-world road conditions, not only good weather but also hazy and rainy weather exist, which can cause serious visual problems. Many studies have been

conducted on removing rain and haze from images, but none have studied removing rain and haze at simultaneously.

On a real road, haze and rain tend to occur simultaneously. However, applying existing algorithms improves only part of the haze or rain. In this paper, the occlusion of haze and rain is both considered and propose a frequency-based haze and rain removal network.

Overall, the framework of our work has three stages. The first is the frequency-based decomposition of the input image using the guided filter, and the second is the proposed FHRR-Net which extracts features for rain streaks and transmission maps to restore images. The network consist two symmetric encoder-decoder networks to train the low- and high-frequency networks separately to remove the rain streaks. Finally, the haze is removed with image degradation model.

Proposed FHRR-Net are compared with two state-of-the-art methods on synthesized road test images and real urban road test images. Better results are shown in test data synthesized through experiments, as well as in real test images, than other state-of-the-art methods. Furthermore, object detection experiments validated that FHRR-Net is helpful for general real-world vision applications.

**Author Contributions:** Conceptualization, D.H.K. and M.T.L.; Data curation, W.J.A. and D.W.K.; Formal analysis, D.H.K., W.J.A., and M.T.L.; Methodology, D.H.K., W.J.A., M.T.L., and T.K.K.; Software, D.H.K., T.K.K., and D.W.K.; Validation, M.T.L. and T.K.K.; Writing—original draft, D.H.K. and M.T.L.; Writing—review & editing, M.T.L. and T.K.K. All authors have read and agreed to the published version of the manuscript.

**Funding:** This research was supported by the Basic Science Research Program through the National Research Foundation of Korea (NRF) funded by the Ministry of Sciences and ICT (Grants No. NRF-2016R1D1A1B01016071 and No. NRF-2019R1A2C1089742).

**Institutional Review Board Statement:** Not applicable.

**Informed Consent Statement:** Not applicable.

**Data Availability Statement:** The Cityscapes data presented in this study are openly available in Cityscapes paper [32].

**Conflicts of Interest:** The authors declare no conflict of interest.

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
