# Peer review of "Frequency-Based Haze and Rain Removal Network (FHRR-Net) with Deep Convolutional Encoder-Decoder"

_applsci, doi:10.3390/app11062873_

Round 1

Reviewer 1 Report

The work developed in this paper presents a clear improvement in the process of image enhancement with fog and rain using guided filters and connectionist computing. 

The topic continues to be of interest in the field of computer vision because of its implications in the domain of autonomous driving.

No inconsistencies or doubts have been found in the methodology, development and tests performed beyond some punctual element, such as the fact that, in addition to including the number and database of images with which the training has been performed, the number of images with which the calibration and test has been performed should also be included.

Another aspect that, eventually, could have been included (although I do not consider it necessary for the quality of the article) is an analysis of the error functions of equations (7) and (8) that would effectively demonstrate that they are in the same range and with similar growths.

A minor issue: check spelling at some points. An error has been found in the text of Figure 9

Author Response

Please refer to Revision Note.

Reviewer 2 Report

This work presents a solution to remove haze and rain in sigle images, using a neural network.

In page 4, line 104 (equation 4), "where O operator indicates....". May be the O should be the operator \circle in previous equation.

All information about experiment design is missing, and it is very important to support conclusions.

  • What databases has been used? How and Why have been selected?
  • Tables 2, 3 and 4 presents different results. How have been computed this values? k-Crossvalidation is mandatory to avoid biased results.
  • Only is presented the average (or only one experiment). It is mandatory to provide mean and variance.
  • It is necessary to carry out a hypothesis test to show that the proposed work is better than others.

Author Response

Please refer to Revision Note. 

Round 2

Reviewer 2 Report

The changes proposed by authors solve the questions.

But, the presentation of mean and variance values is not appropiated. The expression 22.02 ±1.20 is an interval, that could be used to define a confidence interval. But I understand that it is not the case. It must be written µ=22.02 σ2=1.20. 

Author Response

Please refer to Revision Note_Reviewer 2_ Round 2.

This manuscript is a resubmission of an earlier submission. The following is a list of the peer review reports and author responses from that submission.

Round 1

Reviewer 1 Report

My main concern about the paper are the experiments and the lack of explanation of the methods where both low and high frequency images are  handled seemingly the same without any explanation. Additionally, the method seems to handle haze and rain the same. In the experiments the method is compared to two completely orthogonal method which claims to handle only haze or rain both none of them both. It is not a fair comparison since the artificial dataset (which is also a problematic point since the authors use a dataset which has at least haze thus they could easily compare their method to MSCNN int he original valid data) seems even on those low (the images are fairly low resolution) quality images fake. The authors should compare their method in a fair comparison (and compare it with a derained+mscnnn or vice versa method) and measure the increment in object detection on a dataset not on two images.   

Author Response

 Please refer to Revision Note.

Reviewer 2 Report

This paper describes a method to mitigate atmospheric effects that degrade image quality in automotive applications.

In general I think the notational accuracy of the equations should be improved, as well as the description of the methods and algorithms implemented. The level of explanatory quality that allows reproducible research is not achieved. I trust this can be improved with a round of review though.

The following is a list of major detail comments and suggestions:

  • The manuscript doesn’t motivate some design choices: the choice for a neural network to dehaze and remove rain streaks is sound, but why start from preprocessed images? A neural network might as well start from the original input image. That said, the separation between high and low frequency content may be done in many many ways, guided filter just being one of them. How is the currently chosen architecture motivated and validated?
  • The streak model for rain is very simplistic, it ignores important aspects like scene depth which surely impact the appearance of rain (distant rain streaks look different from closeby streaks). Furthermore, the blending approach is not motivated other than in the work of Luo [22], while I am convinced that this blending approach is more suitable than linear addition, both this work nor the work of Luo explains this blending approach from the physical modelling perspective. This is a missed opportunity. In fact, the Wikipedia page is a bit more revealing although doesn’t talk about physics explicitly: https://en.wikipedia.org/wiki/Blend_modes#:~:text=With%20Screen%20blend%20mode%20the,is%20the%20top%20layer%20value.
  • Many of the resulting images are anecdotical (very few samples), which is not convincing due to the possibility of cherry picking results. Furthermore figures 3, 9, 10 are too small to allow the reader to evaluate performance on individual rain streaks. These figures should be cropped to be more useful. Also, in general, more data needs to be processed using better-considered metrics to convincingly show the value of the proposed technique.
  • What is Fu negative mapping method? This is inadequately described.
  • Equation 9 is proposed as the estimate of J from equation 1. First of all, I would urge the authors to pursue notational consistency (in general, not just with respect to these two equations). Furthermore, as an estimator this is poorly conditioned due to lack of actual statistical modelling. For example: what happens if t(x)=0, which is a real possibility? This is covered by the arbitrary threshold later, but this is a rather un-elegant solution that may introduce artifacts.
  • In section 3.3, a performance comparison to SOTA methods is performed. This was performed on synthetic datasets. A number of crucial details are missing. For example, in what sense were these synthetic datasets different from the training set? Furthermore, I don’t think it is a fair comparison to compare quantitively to methods that may be trained on different type of datasets, as utilizing the same synthetic dataset generation procedure for training and testing gives an unfair and unrealistic advantage to the proposed method. The authors even acknowledge this unwittily, by remarking that the other methods are even built for rain streak removal (like MSCNN). Comparing such methods quantitatively in terms of pixel metrics like PSNR is thus not useful.
  • The method relies heavily on synthetic data, yet the paper does not attempt to validate the technique to synthesize such data. The danger with this way of working is that major realistic effects are ignored, which may result in the method underperforming on real data. Figure 9 is therefore interesting, as it is a rare example of the method being validated on real-world examples. Unfortunately, the few images shown there are too small to analyse. Furthermore, the sample size (only 4 images) too low to derive thorough conclusions from.
  • The idea to validate the proposed approach by evaluating object detection performance in real-world conditions is sound and I would encourage the authors to pursue it. However in its current state, evaluating detection performance impact by calculating pixel-level features on a small number of cars (table 3) is not a convincing way to do it. The number of images and cars is far too low to derive statistically significant conclusions from. A better method is to evaluate detection performance in terms of precision-recall or average precision by treating the yolo output as binary detection results. Thus the impact of the proposed method may be evaluated on thousands of cars rather than a cherry-picked subset of very few images.

Here is a list of minor comments:

  • The manuscript states: “Recently, deep learning-based methods have been studied to solve the problem of a lack of information in a single image” -> This doesn’t make sense, deep learning actually suffers from a lack of ground truth? Or do the authors mean that deep learning, contrary to previous machine learning methods learns features from many images, not just based on a model trained on one image?
  • The manuscript states “This phenomenon is due to the convergence problem characteristics:L2 tends to converge into to the local minimum, but L1 converges well to the global minimum in imagerestoration” -> This is not true. L1 simply corresponds better to human visual acuity to spot visible differences.
  • The manuscript uses superfluous and needlessly complex notational conventions: Why is multiplication in equation 4 written using an asterisk symbol? These equations could just as well be described as functions, which would make the notations consistent with the rest of the paper.
  • The description in section 2.3.3 is incomplete and vague. What does “a ground truth image where only haze exists” mean? What is Xi? How is a transmission map defined? Why the notational inconsistency with equation (2)?
  • The description of the experimental environment is unclear. For example: “We set the learning rate to 0.01for 100K and 0.001 for 100K to 200K and terminate the training at ” What is K ? 1000 epochs? 1000 iterations?

Author Response

Please refer to Revision Note.
